# A Relation-Theoretic Formulation of Browder–Göhde Fixed Point Theorem

Aftab Alam [1], Mohammad Imdad [1], Mohammad Asim [2,*] and Salvatore Sessa [3,*]

1   Department of Mathematics, Aligarh Muslim University, Aligarh 202002 , India; aafu.amu@gmail.com (A.A.); mhimdad@gmail.com (M.I.)
2   Department of Mathematics, Shree Guru Gobind Singh Tricentenary University, Gurugram 122505, India
3   Dipartimento di Architettura, Università degli Studi di Napoli Federico II, 80138 Napoli, Italy
*   Correspondence: mailtoasim27@gmail.com and sessa@unina.it

**Abstract:** In this paper, we introduce the concept of $\mathcal{R}$-nonexpansive self-mappings defined on a suitable subset $K$ of a Banach space, wherein $\mathcal{R}$ stands for a transitive binary relation on $K$, and utilize the same to prove a relation-theoretic variant of classical Browder–Göhde fixed point theorem. As consequences of our newly proved results, we are able to derive several core fixed-point theorems existing in the literature.

**Keywords:** binary relation; $\mathcal{R}$-nonexpansive mappings; uniformly convex banach space; $\mathcal{R}$-intervals





## 1. Introduction

Metric fixed-point theory is a relatively old but still young area of research which occupies an important place in nonlinear functional analysis. In fact, the strength of fixed-point theory lies in it's wide range of applications, which include optimization theory, approximation theory, variational inequalities, economic theory, operator equations, fractal theory, control theory, global analysis, physics, statistics, engineering, computer science, biology, chemistry and several others. There exists extensive literature on this topic, which includes pure as well as applied aspects. Indeed, the most popular result of metric fixed point theory is the classical Banach contraction principle, which is essentially due to S. Banach [1] (proved in 1922). Many authors extended the Banach contraction principle employing relatively more general contractive conditions, e.g., see [2–12] and references therein.

In 2004, Ran and Reurings [13] extended the Banach contraction principle to a metric space endowed with a partial order and utilized the same for solving some special matrix equations. Thereafter, Nieto and Rodríguez-López [14] slightly modified the Ran–Reurings theorem and gave the application to solve boundary value problems in differential equations. Here it can be pointed out that the fixed point results of Ran and Reurings [13] and Nieto and Rodríguez-López [14] are consequences of the results of Turinici [15,16] proved in 1986 (for further details, we refer [17,18]). In the same continuation, Alam and Imdad [19] obtained a variant of the Banach contraction principle under an arbitrary binary relation (not necessarily a partial order).

On the other hand, the origin of metric fixed point theory for nonexpansive mappings on Banach spaces can be traced back to four simultaneous articles of Browder [20,21], Göhde [22] and Kirk [23]. In order to ascertain the existence of a fixed point of a nonexpansive mapping, it is necessary for the underlying Banach space to own specific geometric properties. Incidentally, Hilbert spaces enjoy such properties, which allows Browder [20] to prove a natural result in Hilbert spaces.

**Theorem 1** (Browder Fixed-Point Theorem [20]). *Let H be a Hilbert space, K a bounded, closed and convex subset of H and $T : K \to K$ a nonexpansive mapping. Then T has a fixed point.*

Generally speaking, all Banach spaces are not equipped with desired geometric properties which are enough to ensure the existence of fixed-point results for nonexpansive mappings. In fact, the class of "uniformly convex Banach spaces" is adjudged suitable to prove fixed-point results for nonexpansive mappings.

**Definition 1.** *A Banach space $(X, \| \cdot \|)$ is called uniformly convex if for every $\varepsilon > 0$ there is a $\delta > 0$, such that for any $x, y \in X$ with*

$$\|x\| \leq 1, \ \|y\| \leq 1, \ \|x - y\| \geq \varepsilon$$

$$\implies \frac{1}{2}\|(x + y)\| \leq 1 - \delta.$$

**Example 1.** *Euclidean space $\mathbb{R}^n$ is uniformly convex with Euclidean norm $\|x\| = \left( \sum\limits_{i=1}^{n} x_i^2 \right)^{1/2}$, while under the norm $\|x\| = \sum\limits_{i=1}^{n} |x_i|$, it is not uniformly convex.*

After appearance of Theorem 1 in 1965, in the same year Browder [21] and Göhde [22] both independently proved a result in the setting of a uniformly convex Banach space.

**Theorem 2** (Browder-Göhde Fixed-Point Theorem [21,22]). *Let X be a uniformly convex Banach space, K a bounded, closed and convex subset of X and $T : K \to K$ a nonexpansive mapping. Then T has a fixed point.*

Here, it can be pointed out that Browder [21] and Göhde [22] employed different arguments in their proofs. In a well-written monograph, Goebel and Reich [24] chose to give three different proofs of Theorem 2; one of the proofs is based on the intersection method, whereas the other two proofs bank on the idea of asymptotic centres. Kirk [23] proved the same result in the setting of reflexive Banach space having a specific property called "Normal Structure".

**Theorem 3** (Kirk Fixed-Point Theorem [23]). *Let X be a reflexive Banach space and K a bounded, closed and convex subset of X. Suppose that K has normal structure and $T : K \to K$ is a nonexpansive mapping. Then T has a fixed point.*

For a technical description of "Normal Structure", one can consult the monograph of Goebel and Kirk [25].

**Remark 1.** *As every Hilbert space is uniformly convex, Theorem 1 follows from Theorem 2. Additionally, since every uniformly Banach space is reflexive and has normal structure, Theorem 2 follows from Theorem 3. Hence, out of all three above-mentioned theorems, Theorem 3 due to Kirk [23] is most general.*

In 2015, Bachar and Khamsi [26] introduced the idea of monotone nonexpansive mapping defined on a Banach space equipped with a partial ordering. Following Bachar and Khamsi [26], given a partially ordered Banach space $(X, \| \cdot \|, \preceq)$, a mapping $T : D(T) \subseteq X \to X$ is called monotone if $T(x) \preceq T(y)$ for all $x, y \in D(T)$ with $x \preceq y$. Additionally, if $\|Tx - Ty\| \leq \|x - y\|$ for all $x, y \in D(T)$ with $x \preceq y$, then $T$ is said to be monotone nonexpansive. For any $a, b \in X$, the subsets $[a, \to) = \{x \in X : a \preceq x\}$ and $(\leftarrow, b] = \{x \in X : x \preceq b\}$ refer to the order intervals in the partially ordered set $(X, \preceq)$ with initial point $a$ and with end point $b$, respectively. For further relevant details on monotone nonexpansive mappings, one can consult [27–33]. Recently, using the Baire's category

approach, Reich and Zaslavski [33] established the fact that the fixed-point problem for a monotone nonexpansive mapping is well-posed. In this continuation, Bin Dehaish and Khamsi [29] proved an order-theoretic analog of Browder-Göhde fixed point theorem (i.e., Theorem 2), which runs as follows:

**Theorem 4** ([29]). *let* $(X, \| \cdot \|, \preceq)$ *be a partially ordered Banach space such that order intervals are convex and closed. Assume X is uniformly convex. Let K be a bounded closed convex nonempty subset of X. Let* $T : K \to K$ *be a monotone nonexpansive mapping. Assume there exists* $x_0 \in K$ *such that* $x_0$ *and* $T(x_0)$ *are comparable. Then T has a fixed point.*

In this paper, we establish an improved version of Theorem 4. Our improvement is four-fold:

- The partial ordering is replaced by a transitive binary relation;
- The transitivity of the relation is not needed on the whole space $X$, but it can be limited on a suitable subset $K$ of $X$;
- The closedness and convexity of all order intervals in $X$ are not required, but it suffices that merely certain relational intervals in $K$ are closed and convex. Furthermore, the convexity and closedness of whole set $K$ are also relaxed;
- The boundedness of the whole set $K$ is replaced by a relatively weaker assumption.

## 2. Relation-Theoretic Notions

In this section, to make our exposition self-contained, we give some definitions in respect of binary relations, which are used to prove our main results. In what follows, $\mathbb{N}$ and $\mathbb{N}_0$ denote the sets of natural numbers and whole numbers, respectively (i.e., $\mathbb{N}_0 = \mathbb{N} \cup \{0\}$). Recall that a binary relation on a nonempty set $K$ is a subset $\mathcal{R}$ of $K^2$ (i.e., $\mathcal{R} \subseteq K^2$). Trivially, $K^2$ and $\varnothing$ being subsets of $K^2$ are binary relations on $K$, which are, respectively, called the universal relation (or full relation) and empty relation.

Throughout this paper, $\mathcal{R}$ stands for a nonempty binary relation, but for the sake of simplicity, we often write 'binary relation' instead of 'nonempty binary relation'.

**Definition 2** ([34]). *Let K be a nonempty set and* $\mathcal{R}$ *a binary relation of K.*

*(1) The inverse or transpose or dual relation of* $\mathcal{R}$, *denoted by* $\mathcal{R}^{-1}$, *is defined by* $\mathcal{R}^{-1} := \{(x, y) \in K^2 : (y, x) \in \mathcal{R}\}$.

*(2) The symmetric closure of* $\mathcal{R}$, *denoted by* $\mathcal{R}^s$, *is defined to be the set* $\mathcal{R} \cup \mathcal{R}^{-1}$ *(i.e.,* $\mathcal{R}^s := \mathcal{R} \cup \mathcal{R}^{-1}$). *Indeed,* $\mathcal{R}^s$ *is the smallest symmetric relation on K containing* $\mathcal{R}$.

**Remark 2.** $\mathcal{R}^{-1}$ *is transitive if* $\mathcal{R}$ *is transitive.*

**Definition 3** ([19]). *Let* $\mathcal{R}$ *be a binary relation on a nonempty set K and* $x, y \in K$. *We say that x and y are* $\mathcal{R}$-*comparative if either* $(x, y) \in \mathcal{R}$ *or* $(y, x) \in \mathcal{R}$. *We denote it by* $[x, y] \in \mathcal{R}$.

**Proposition 1** ([19]). *For a binary relation* $\mathcal{R}$ *on a nonempty set K,*

$$(x, y) \in \mathcal{R}^s \iff [x, y] \in \mathcal{R}.$$

**Definition 4** ([19]). *Let K be a nonempty set and* $\mathcal{R}$ *a binary relation on K. A sequence* $\{x_n\} \subset K$ *is called* $\mathcal{R}$-*preserving if*

$$(x_n, x_{n+1}) \in \mathcal{R} \quad \forall n \in \mathbb{N}_0.$$

*Also, the sequence* $\{x_n\}$ *is called* $\mathcal{R}$-*reversing if*

$$(x_{n+1}, x_n) \in \mathcal{R} \quad \forall n \in \mathbb{N}_0.$$

**Definition 5** ([19])**.** *Let K be a nonempty set and $T : K \to K$ a mapping. A binary relation $\mathcal{R}$ on K is called T-closed if for all $x, y \in K$,*

$$(x, y) \in \mathcal{R} \Rightarrow (Tx, Ty) \in \mathcal{R}.$$

**Proposition 2** ([35])**.** *Let K be a nonempty set, $\mathcal{R}$ a binary relation on K and T a self-mapping on K. If $\mathcal{R}$ is T-closed, then, for all $n \in \mathbb{N}_0$, $\mathcal{R}$ is also $T^n$-closed, where $T^n$ denotes the nth iterate of T.*

Now, we present the relation-theoretic variants of monotone nonexpansive mappings and order intervals.

**Definition 6.** *Let K be a subset of Banach space $(X, \| \cdot \|)$ and $\mathcal{R}$ a binary relation of K. A mapping $T : K \to K$ is called $\mathcal{R}$-nonexpansive if*

(i)     *$\mathcal{R}$ is T-closed;*
(ii)    *for all $x, y \in K$ with $(x, y) \in \mathcal{R}$,*

$$\|Tx - Ty\| \leq \|x - y\|.$$

**Remark 3.** *The following conclusions are straightforward.*

(i)     *T is $\mathcal{R}$-nonexpansive $\iff$ T is $\mathcal{R}^{-1}$-nonexpansive.*
(ii)    *T is $\mathcal{R}$-nonexpansive $\iff$ T is $\mathcal{R}^s$-nonexpansive.*
(iii)   *Under universal relation $\mathcal{R} = X^2$, the notion of $\mathcal{R}$-nonexpansive mapping reduces to that of nonexpansive mapping.*

**Definition 7.** *Given a binary relation $\mathcal{R}$ on a nonempty set K, the image of an element $a \in K$ (under the binary relation $\mathcal{R}$) or $\mathcal{R}$-interval with initial point $a \in K$ is a subset of K defined by*

$$\text{Im}(a, \mathcal{R}) = \{x \in K : (a, x) \in \mathcal{R} \text{ or } x = a\}.$$

*Similarly, the preimage of $a \in K$ or $\mathcal{R}$-interval with end point $a \in K$ is a subset of K defined by*

$$\text{PreIm}(a, \mathcal{R}) = \{x \in K : (x, a) \in \mathcal{R} \text{ or } x = a\}.$$

**Remark 4.** *The following conclusions are immediate.*

$$\text{Im}(a, \mathcal{R}) = \text{PreIm}(a, \mathcal{R}^{-1}),$$

$$\text{PreIm}(a, \mathcal{R}) = \text{Im}(a, \mathcal{R}^{-1}),$$

$$\text{Im}(a, \mathcal{R}^s) = \text{PreIm}(a, \mathcal{R}^s).$$

**Remark 5.** *Under $\mathcal{R} := \preceq$, a partial ordering, $\text{Im}(a, \mathcal{R})$ and $\text{PreIm}(a, \mathcal{R})$ coincide with order intervals $[a, \to)$ and $(\leftarrow, a]$ respectively.*

In a Banach lattice, $\mathcal{R}$-intervals are closed and convex under $\mathcal{R} := \preceq$ (cf. [36]).

## 3. Main Results

For the sake of self-containment, we recall the following well-known results due to Smulian [37], which characterizes the reflexivity of Banach space.

**Lemma 1** ([37])**.** *A Banach space X is reflexive if every decreasing sequence $\{K_n\}$ of nonempty bounded, closed and convex subsets of X has nonempty intersections, i.e., $\bigcap_{n=0}^{\infty} K_n \neq \emptyset$.*

**Definition 8** ([29]). *Let* $(X, \|\cdot\|)$ *be a Banach space and* $K \subseteq X$ *and* $\{x_n\}$ *a bounded sequence in K. Then, a function* $\tau : K \to [0, \infty)$ *defined by*

$$\tau(x) = \limsup_{n \to \infty} \|x_n - x\| \quad \forall x \in K,$$

*is called a type function generated by* $\{x_n\}$.

**Lemma 2** ([29]). *Let K be a nonempty, closed and convex subset of a uniformly convex Banach space* $(X, \|\cdot\|)$ *and* $\tau : K \to [0, \infty)$ *a type function. Then,* $\tau$ *has a unique minimum point* $z \in K$, *such that*

$$\tau(z) = \inf\{\tau(x) : x \in K\}.$$

Now, we present a variant of Browder-Göhde fixed-point theorem under a transitive binary relation, which improves Theorem 4.

**Theorem 5.** *Let* $(X, \|\cdot\|)$ *be a uniformly convex Banach space and K a nonempty subset of X. Let* $\mathcal{R}$ *be a transitive binary relation on K and* $T : K \to K$ *an* $\mathcal{R}$-*nonexpansive mapping. If there exists* $x_0 \in K$ *such that*

*(a)* $(x_0, Tx_0) \in \mathcal{R}$;
*(b)* $\mathrm{Im}(T^n x_0, \mathcal{R})$ *is nonempty, closed and convex for each* $n \in \mathbb{N}_0$;
*(c)* $\{T^n x_0\}$ *is bounded;*

*then T has a fixed point.*

**Proof.** Based at the initial point $x_0 \in K$, we can define a Picard sequence $\{x_n\} \subset K$ as follows:

$$x_n := T^n(x_0), \forall\, n \in \mathbb{N}. \tag{1}$$

By assumption $(a)$, we have $(x_0, Tx_0) \in \mathcal{R}$. Hence, using $T$-closedness of $\mathcal{R}$ and Proposition 2, we obtain

$$(T^n x_0, T^{n+1} x_0) \in \mathcal{R},$$

which by using (1), reduces to

$$(x_n, x_{n+1}) \in \mathcal{R} \quad \forall n \in \mathbb{N}_0. \tag{2}$$

Thus, the sequence $\{x_n\}$ is $\mathcal{R}$-preserving.

In view of assumption $(b)$, for each $n \in \mathbb{N}_0$, $M_n := \mathrm{Im}(x_n, \mathcal{R})$ is a closed and convex subset of $X$. Then, $\{M_n\}$ is bounded as $\{x_n\}$ is bounded (by assumption $(c)$). Clearly, $\{M_n\}$ is decreasing, *i.e.*, $M_n \supseteq M_{n+1}$ for all $n \in \mathbb{N}_0$. Hence, $\{M_n\}$ is a bounded decreasing sequence of closed and convex subsets of $X$. Also, the Banach space $X$ being uniformly convex is reflexive, so using Lemma 1, we have

$$M := \bigcap_{n=0}^{\infty} \mathrm{Im}(x_n, \mathcal{R}) \neq \emptyset.$$

Take $x \in M$; then $(x_n, x) \in \mathcal{R}$ for each $n \in \mathbb{N}_0$. Using the $\mathcal{R}$-closedness of $T$, we have

$$(Tx_n, Tx) \in \mathcal{R},$$

which by using (1), reduces to

$$(x_{n+1}, Tx) \in \mathcal{R} \quad \forall n \in \mathbb{N}_0. \tag{3}$$

Using (2), (3) and transitivity of $\mathcal{R}$, we get

$$(x_n, Tx) \in \mathcal{R},$$

which yields that

$$T(M) \subseteq M. \tag{4}$$

In lieu of assumption $(c)$, $\{x_n\}$ is bounded. Therefore, consider the type function $\tau : K \to [0, \infty)$ generated by $\{x_n\}$, *i.e.*,

$$\tau(x) = \limsup_{n \to \infty} \|x_n - x\|. \tag{5}$$

By Lemma 2, there exists a unique minimum point $z \in M$, *i.e.*,

$$\tau(z) = \inf_{x \in M} \tau(x).$$

Now, using (4) and (5), we obtain

$$\tau(Tz) = \limsup_{n \to \infty} \|x_n - Tz\| = \limsup_{n \to \infty} \|Tx_n - Tz\|. \tag{6}$$

As $z \in M$, we have $(x_n, z) \in \mathcal{R}$. Hence, by the $T$-closeness of $\mathcal{R}$, we obtain

$$(Tx_n, Tz) \in \mathcal{R}. \tag{7}$$

Using (7) and the $\mathcal{R}$-nonexpansiveness of $T$, we get

$$\|Tx_n - Tz\| \leq \|x_n - z\|. \tag{8}$$

Using (6) and (8), we get

$$\tau(Tz) \leq \limsup_{n \to \infty} \|x_n - z\| = \tau(z).$$

By the uniqueness of the minimum point of $\tau$, we get $T(z) = z$ so that $z$ is a fixed point of $T$. $\square$

Taking $\mathcal{R}^{-1}$ instead of $\mathcal{R}$ in Theorem 5 and using Remarks 2–4, we get the following dual result.

**Theorem 6.** *Let $(X, \|\cdot\|)$ be a uniformly convex Banach space and $K$ a nonempty subset of $X$. Let $\mathcal{R}$ be a transitive binary relation on $K$ and $T : K \to K$ an $\mathcal{R}$-nonexpansive mapping. If there exists $x_0 \in K$ such that*

$(a)$    $(Tx_0, x_0) \in \mathcal{R}$;
$(b)$    $\mathrm{PreIm}(T^n x_0, \mathcal{R})$ *is nonempty, closed and convex for each $n \in \mathbb{N}_0$*;
$(c)$    $\{T^n x_0\}$ *is bounded*;
*then $T$ has a fixed point.*

## 4. Certain Consequences

In this section, we point out that several core fixed-point theorems turn out to be consequences of our newly proved results. Under the universal relation $\mathcal{R} = X^2$, Theorem 5 (similarly, Theorem 6) reduces the classical Browder-Göhde fixed point theorem (i.e., Theorem 2). Observe that in the presence of universal relation, the whole set $K$ is closed, convex and bounded. Consequently, every arbitrary $x_0 \in K$ satisfies the assumptions $(a)$, $(b)$ and $(c)$.

Choosing $\mathcal{R}$ to a partial order $\preceq$ in Theorems 5 and 6, we obtain the sharpened versions of Theorem 4 established by Bin Dehaish and Khamsi [29] in the forms of the following two corollaries:

**Corollary 1.** *Let* $(X, \| \cdot \|)$ *be a uniformly convex Banach space and K a nonempty subset of X. Let* $\preceq$ *be a partial order on K and* $T : K \to K$ *a monotone nonexpansive mapping. If there exists* $x_0 \in K$ *such that*

*(a)* $x_0 \preceq T(x_0)$;
*(b) for each* $n \in \mathbb{N}_0$, *the order interval* $[T^n x_0, \to)$ *is nonempty, closed and convex;*
*(c)* $\{T^n x_0\}$ *is bounded;*
*then T has a fixed point.*

**Corollary 2.** *Let* $(X, \| \cdot \|)$ *be a uniformly convex Banach space and K a nonempty subset of X. Let* $\preceq$ *be a partial order on K and* $T : K \to K$ *a monotone nonexpansive mapping. If there exists* $x_0 \in K$ *such that*

*(a)* $T(x_0) \preceq x_0$;
*(b) for each* $n \in \mathbb{N}_0$, *the order interval* $(\leftarrow, T^n x_0]$ *is nonempty, closed and convex;*
*(c)* $\{T^n x_0\}$ *is bounded;*
*then T has a fixed point.*

Recall that a nonempty and nontrivial subset $P$ of a real Banach space $(X, \| \cdot \|)$ is said to be a cone if $\alpha P \subset P$ for all $\alpha \geq 0$ and $P \cap (-P) = \{0\}$. On $X$, we define a partial ordering $\preceq$ with respect to $P$ as follows:

$$x \preceq y \iff y - x \in P \quad \forall \, x, y \in X.$$

Such a Banach space $(X, \| \cdot \|, \preceq)$ remains an ordered Banach space induced by the cone $P$.

Now, we are equipped to deduce the following two recent results of Song et al. [38] from Corollaries 1 and 2, respectively.

**Corollary 3** ([38])**.** *Let* $(X, \| \cdot \|, \preceq)$ *be a uniformly convex ordered Banach space induced by a closed convex cone P and K a nonempty and closed convex subset of X. Also, suppose that* $T : K \to K$ *is a monotone nonexpansive mapping. If there exists* $x_0 \in K$ *such that* $x_0 \preceq Tx_0$ *and the sequence* $\{T^n x_0\}$ *is bounded, then T has a fixed point.*

**Proof.** We show that for each $n \in \mathbb{N}_0$, the order interval $[T^n x_0, \to)$ is closed and convex. Take $z_1, z_2 \in [T^n x_0, \to)$; then $z_1 - T^n(x_0) \in P$ and $z_2 - T^n(x_0) \in P$. Hence, for $\lambda \in (0, 1)$, by convexity of $P$, we have

$$\lambda z_1 + (1 - \lambda) z_2 - T^n(x_0) = \lambda(z_1 - T^n x_0) + (1 - \lambda)(z_2 - T^n x_0) \in P$$

implying thereby $\lambda z_1 + (1 - \lambda) z_2 \in [T^n x_0, \to)$. Therefore, $[T^n x_0, \to)$ is convex. Now, take $\{z_m\} \subset [T^n x_0, \to)$ such that $\lim_{m \to \infty} z_m = z$. Then for each $m \in \mathbb{N}$, we have $z_m - T^n(x_0) \in P$. Hence, using the closedness of $P$, we have

$$z - T^n(x_0) = \lim_{m \to \infty} (z_m - T^n x_0) \in P$$

which yields that $z \in [T^n x_0, \to)$. Thus, $[T^n x_0, \to)$ is closed. Therefore, the assumption $(b)$ of Corollary 1 holds. Consequently, our result is deducible from Corollary 1. $\square$

**Corollary 4** ([38])**.** *Let* $(X, \| \cdot \|, \preceq)$ *be a uniformly convex ordered Banach space induced by a closed convex cone P and K a nonempty and closed convex subset of X. Also, suppose that* $T : K \to K$ *is a monotone nonexpansive mapping. If there exists* $x_0 \in K$ *such that* $Tx_0 \preceq x_0$, *the sequence* $\{T^n x_0\}$ *is bounded, then T has a fixed point.*

**Proof.** Using the arguments similar to the lines of the proof of Corollary 3, one can show that for each $n \in \mathbb{N}_0$, the order interval $(\leftarrow, T^n x_0]$ is closed and convex. Thus, all the conditions of the hypotheses of Corollary 3 are satisfied and hence the conclusion is immediate. □

**Author Contributions:** All the authors have contributed equally in all parts. All authors have read and agreed to the published version of the manuscript.

**Funding:** This research received no external funding.

**Acknowledgments:** The authors are thankful to two learned referees for their fruitful suggestions and constructive comments towards the improvement of this paper.

**Conflicts of Interest:** The authors declare no conflict of interest.

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
