# Peer review of "A Relation-Theoretic Formulation of Browder–Göhde Fixed Point Theorem"

_axioms, doi:10.3390/axioms10040285_

Round 1
Reviewer 1 Report
The authors introduced a class of binary relation type nonexpansive elf mapping and established some fixed point results for this class of mappings in the Banach space settings. The results obtained are new and correct as well as nontrivial extension of several classical results. This paper surely deserves publication. But it needs revision. Typos should be checked carefully and corrected. For example, in Definition 8 and Lemma 2, "x" should be "r". Also the following very recent and closely related papers are requested to be mentioned and cited to make the References more complete. (1) Mahtab Delfani, Ali Farajzadeh, Ching-Feng Wen, Some fixed point theorems of generalized -contraction mappings in -metric spaces, J. Nonlinear Var. Anal. 5 (2021), 615-625. (2) Tomonari Suzuki, Edelstein’s fixed point theorem in semimetric spaces, J. Nonlinear Var. Anal. 2 (2018), 165-175.
Author Response
|
|
Referee’s comments |
Author’s improvement |
|
Referee-1 |
The following very recent and closely related papers are requested to be mentioned and cited to make the References more complete. (1) Mahtab Delfani, Ali Farajzadeh, Ching-Feng Wen, Some fixed point theorems of generalized -contraction mappings in -metric spaces, J. Nonlinear Var. Anal. 5 (2021), 615-625. (2) Tomonari Suzuki, Edelstein’s fixed point theorem in semimetric spaces, J. Nonlinear Var. Anal. 2 (2018), 165-175. |
The suggestion is incorporated by adding the papers in references and citing them.
|
|
In Definition 8 and Lemma 2, "x" should be "r". |
All such errors are correcting by taking similar elements x, y everywhere in the entire manuscript. |
|
|
Referee-2 |
page 4, Definition 7, replace x with r. |
|
|
page 4, Definition 8, replace x with r.
|
||
|
page 4, Lemma 2, replace x with r.
|
||
|
Referee-2 |
Page 1, line 9, there are some terms like origination which can be replaced |
The suggestion is incorporated replacing the term ‘origination’ with ‘origin’ |
|
Page 3, line 74, replace utilized with used. |
The suggestion is incorporated replacing the term ‘utilized’ with ‘used’ |
|
|
Page 4, line 105, replace ‘record’ with ‘recall’ and ‘results’ with ‘result’ |
The suggestion is incorporated replacing the term ‘record’ with ‘recall’ and ‘results’ with ‘result’ |
|
|
Page 5, in the proof of Theorem 5, after (3), you wrote...and transitivity of T. Replace T by R. After (4), replace in lieu of assumption with From. |
The suggestion is incorporated by correcting errors |
|
|
Page 7, please reorganize the proof of Corollary 3, by showing first that the order interval is closed and convex and concluding that by applying Corollary 1, the proof is complete. |
The suggestion is incorporated by reorganizing the proof. |

Reviewer 2 Report
My comments are presented below

Author Response

(The authors gave the same response as above.)
